# Agnostic Learning with Multiple Objectives

**Corinna Cortes**
Google Research
New York, NY 10011
corinna@google.com

**Javier Gonzalvo**
Google Research
New York, NY 10011
xavigonzalvo@google.com

**Mehryar Mohri**
Google & Courant Institute
New York, NY 10012
mohri@google.com

**Dmitry Storcheus**
Courant Institute & Google
New York, NY 10012
dstorcheus@google.com

## Abstract

Most machine learning tasks are inherently multi-objective. This means that the learner has to come up with a model that performs well across a number of base objectives $\mathcal{L}_1, \ldots, \mathcal{L}_p$, as opposed to a single one. Since optimizing with respect to multiple objectives at the same time is often computationally expensive, the base objectives are often combined in an ensemble $\sum_{k=1}^{p} \lambda_k \mathcal{L}_k$, thereby reducing the problem to scalar optimization. The mixture weights $\lambda_k$ are set to uniform or some other fixed distribution, based on the learner's preferences. We argue that learning with a fixed distribution on the mixture weights runs the risk of overfitting to some individual objectives and significantly harming others, despite performing well on an entire ensemble. Moreover, in reality, the true preferences of a learner across multiple objectives are often unknown or hard to express as a specific distribution. Instead, we propose a new framework of *Agnostic Learning with Multiple Objectives* (ALMO), where a model is optimized for *any* weights in the mixture of base objectives. We present data-dependent Rademacher complexity guarantees for learning in the ALMO framework, which are used to guide a scalable optimization algorithm and the corresponding regularization. We present convergence guarantees for this algorithm, assuming convexity of the loss functions and the underlying hypothesis space. We further implement the algorithm in a popular symbolic gradient computation framework and empirically demonstrate on a number of datasets the benefits of ALMO framework versus learning with a fixed mixture weights distribution.

## 1 Motivation

Machine learning is inherently a multi-objective task [Jin, 2006, Jin and Sendhoff, 2008]. In most real-world problems, a learner is required to find a model with strong performance across multiple objectives $\mathcal{L}_1, \ldots, \mathcal{L}_p$. Committing to a single objective $\mathcal{L}_k$ often fails to capture the full complexity of the underlying problem and causes models to overfit to that individual objective [Kendall et al., 2018].

For example, the BLEU score [Papineni et al., 2002] is commonly selected as a single objective for model training and evaluation in machine translation literature. However, it *overfits* to short sentences, because it is focused on word-based $n$-gram precision [Duh et al., 2012]. Thus, it is often agreed upon in the research community that machine translation models need to perform well not just in terms of the BLEU score, but also for other metrics that measure various aspects of translation

beyond the $n$-gram similarity, for example, METEOR [Lavie and Agarwal, 2007], which measures synonym matching, or RIBES [Isozaki et al., 2010] which accounts for deviation in word order.

A well-studied approach to working with multiple correlated objectives such as those just discussed is to estimate a Pareto-efficient frontier and seek a solution that lies on that frontier [Jin and Sendhoff, 2008, Sener and Koltun, 2018, Shah and Ghahramani, 2016, Marler and Arora, 2004]. While this approach can provide a rigorous description of the trade-offs between multiple correlated objectives, the estimation of the Pareto-efficient frontier is computationally expensive [Duh et al., 2012, Godfrey et al., 2007] and scales poorly with the number of base objectives. Moreover, optimization along the Pareto-efficient frontier typically requires computing multiple gradients for each step, which is challenging to adapt to large-scale problems. Additionally, even when the Pareto frontier is computed, some selection criterion is needed to choose a point on that frontier. In Section 3, we prove that our solution is actually guaranteed to be Pareto-optimal, thus lying on the frontier. Our algorithms hence provides a theoretical justification for selecting one of the possibly many points on the frontier.

In short, despite the multi-objective nature of machine learning, working with a vector of multiple objectives at the same time turns out to be computationally challenging. For that reason, researchers often combine the set of base objectives $\{\mathcal{L}_1, \ldots, \mathcal{L}_p\}$ into a weighted ensemble $\mathcal{L}_\lambda = \sum_{k=1}^{p} \lambda_k \mathcal{L}_k$ for some $\lambda$ in the simplex $\Delta_p$, essentially making it a scalar function [Jin, 2006]. This allows one to make use of efficient scalar function optimization algorithms on large-scale problems, typically using various stochastic gradient descent techniques [Bottou et al., 2018]. However, working with an ensemble of base objectives raises a natural question: how should we set the mixture weights $\lambda_1, \ldots, \lambda_p$ ? For example, if we define $\mathcal{L}_\lambda = \lambda_1 \cdot \text{BLEU} + \lambda_2 \cdot \text{METEOR}$, should we assume that BLEU and METEOR are equally important and set the mixture weights to be uniform $\lambda_1 = \lambda_2 = 1/2$? Or should we say that METEOR is significantly more important than BLEU for a particular problem and $\lambda_2$ should be set higher that $\lambda_1$?

Despite the simplicity of the questions above, there is no clear answer to how to determine the mixture weights for multi-objective problems. Particularly, this is because there is no straightforward way to map the requirements of a particular problem that a learner is seeking to solve to a corresponding distribution of the mixture weights [Van Moffaert et al., 2014]. Thus, the mixture weights are usually assigned to uniform by default. However, in many cases the uniform ensemble of objectives can do more harm than good. This is simply because fitting models with the uniform mixture weights can significantly hurt some of the individual objectives in the mixture. For many machine learning applications (e.g., vision or speech), a significant decrease in performance on one objective is intolerable even if the performance on the uniform average of objectives is improved.

One can argue that if the uniform combination of objectives is not the natural target, then we should just set mixture weights for every problem separately based on the relative importance of each of the base losses. However, this is still not satisfactory, because the learner's preferences are often shifting over time, they may be unobservable in some cases or even unknown. It is also often the case in the machine learning industry that when several parties develop a model for a particular multi-objective problem, their preferences for the base objectives are conflicting with each other.

Given these challenges of working with a fixed mixture-weight distribution, we argue that the natural goal of a learner with multiple objectives $\{\mathcal{L}_1, \ldots, \mathcal{L}_p\}$ has an *agnostic* nature. That is, a learner wishes to obtain a model that performs well for *any* possible combination of mixture weights in the ensemble $\mathcal{L}_\lambda = \sum_{k=1}^{p} \lambda_k \mathcal{L}_k$. Such a goal is also *risk-averse*, since a model that is robust against any combination of ensemble weights is also robust against their worst (adversarial) combination.

Thus, we propose a new framework of *Agnostic Learning with Multiple Objectives* (ALMO) inspired by the Agnostic Federated Learning algorithm [Mohri et al., 2019], where the underlying model is optimized for any possible distribution of the mixture weights in the ensemble $\mathcal{L}_\lambda = \sum_{k=1}^{p} \lambda_k \mathcal{L}_k$. Instead of optimizing the model for a fixed distribution of $\lambda_1, \ldots, \lambda_p$, with the high risk of over-fitting to a subset of the base objectives, we define the *agnostic* loss function that ensures that the model performs well against any mixture, including the worst-case mixture values. Thus, the ALMO framework closely matches the learning setting used by most real-world ML practitioners, who often seek to deploy a model that is robust against any mixture of base objectives and that does not need to be frequently retrained when the base objective preferences shift.

We give data-dependent generalization bounds based on the Rademacher complexity of the underlying loss functions, which are then used to define the optimization algorithm and the regularization for the

problem. We suggest an efficient optimization algorithm for the ALMO setting that is inspired by the generalization bounds derived in this paper. The algorithm can be applied in a straightforward manner across multiple tasks, since it is based on stochastic gradient descent and can be implemented in popular symbolic gradient computation platforms, including TENSORFLOW [Abadi et al., 2016], KERAS [Chollet et al., 2015] or PYTORCH [Paszke et al., 2019].

We conducted a series of experiments on several datasets that demonstrate the benefits of the ALMO framework versus learning with a uniform mixture weights distribution. The experiments also show that the ALMO framework provides a generally robust model that performs better than baselines on metrics that are not even included in the original set of base objectives. Moreover, while the algorithm's convergence bound holds for convex hypothesis spaces, we show that the ALMO framework provides robust models even for non-convex hypothesis classes, such as Deep Neural Networks.

The rest of the paper is structured as follows. In Section 2, we formally describe the ALMO framework, define the agnostic multi-objective loss function, and discuss the connection of our solution to the Pareto-optimal frontier. In Section 3, we derive the sample-dependent Rademacher complexity bounds for the ALMO framework. In Section 4, we put forward a stochastic gradient-descent-based algorithm inspired by the generalization bounds. In Section 5, we describe our experimental setting and provide empirical results.

## 2   Learning Scenario

In this section, we introduce the formal learning scenario of the ALMO framework. We define the *agnostic* loss function and argue that by optimizing this loss, the learner obtains a model that is robust against any mixture weights distribution in the ensemble $\mathcal{L}_\lambda = \sum_{k=1}^p \lambda_k \mathcal{L}_k$.

We consider the supervised learning setting, in which the learner receives a labeled sample $S = ((x_1, y_1), \ldots, (x_m, y_m))$ drawn i.i.d. from some distribution $\mathcal{D}$ over $\mathcal{X} \times \mathcal{Y}$, where $\mathcal{X}$ denotes the input space and $\mathcal{Y}$ denotes the output space. Let $\mathcal{H}$ be the hypothesis space and $l \colon \mathcal{Y} \times \mathcal{Y} \mapsto \mathbb{R}_+$ a loss function. The loss of $h \in \mathcal{H}$ for a labeled instance $(x, y)$ is $\ell(h(x), y)$, and its expected loss $\mathcal{L}(h) = \mathbb{E}_{(x,y)\sim\mathcal{D}}[\ell(h(x), y)]$. In a standard supervised learning scenario, the goal of the learner is to find a hypothesis with small expected loss.

However, as discussed in the previous section, the majority of learning applications require optimizing over a number of objectives (loss functions) at the same time. In such cases, the learner has access to a set of $p$ *base loss functions* $\{l_1, \ldots, l_p\}$. For any $h \in \mathcal{H}$, these base loss functions have expectations $\{\mathcal{L}_1(h), \ldots, \mathcal{L}_p(h)\}$, where $\forall k \colon \mathcal{L}_k(h) = \mathbb{E}_{(x,y)\sim\mathcal{D}}[\ell_k(h(x), y)]$. The goal of the learner is to obtain a hypothesis $h \in \mathcal{H}$ that performs well across any weighted combination of the expected base losses $\mathcal{L}_\lambda(h) = \sum_{k=1}^p \lambda_k \mathcal{L}_k(h)$, for some $\lambda \in \Delta_p$.

In this paper, we focus on a scenario where the base loss functions come from different functional families. However, another scenario where the base losses represent the application of a single loss function on different output domains i.e. *multi-task learning* is also relevant to our work. Particularly, Kendall et al. [2018] work in a setting where the base losses are the applications of Gaussian likelihood on distinct domains. Thus, they learn the $1/\sigma$ parameter as mixture weights in the linear ensemble of Gaussian densities. The work of Chen et al. [2018] points out that when the same objective function is applied to different domains, it can have different magnitudes, which is a challenge for gradient-based optimization methods. They suggest the GRADNORM algorithm that dynamically scales gradients in different output subspaces. GRADNORM can be used as a subroutine in our optimization problem, however, directly normalizing the base losses was sufficient for experiments in this paper.

We assume that the learner has a defined boundary set of preferences over the mixture of objectives that can be described by a convex subset $\Lambda \subseteq \Delta_p$. However, the true mixture coefficients $\lambda$ within $\Lambda$ are unknown. Thus, the learner seeks a solution that is favorable for any possible $\lambda \in \Lambda$. Therefore, we define the *agnostic multi-objective loss* $\mathcal{L}_\Lambda(h)$ associated to a predictor $h \in \mathcal{H}$ and a subset $\Lambda \subseteq \Delta_p$ as follows:

$$\mathcal{L}_\Lambda(h) = \max_{\lambda \in \Lambda} \mathcal{L}_\lambda(h). \tag{1}$$

The goal of the learner is to find $h_\Lambda \in \mathcal{H}$ solution of the following optimization problem:

$$\min_{h \in \mathcal{H}} \mathcal{L}_\Lambda(h). \tag{2}$$

To determine the best hypothesis, a sample estimate of the agnostic multi-objective loss $\mathcal{L}_\Lambda(h)$ is needed. Thus, we define the *empirical agnostic multi-objective loss* $\widehat{\mathcal{L}}_\Lambda(h)$ as follows:

$$\widehat{\mathcal{L}}_\Lambda(h) = \max_{\lambda \in \Lambda} \sum_{k=1}^{p} \lambda_k \widehat{\mathcal{L}}_k(h) = \max_{\lambda \in \Lambda} \sum_{k=1}^{p} \lambda_k \frac{1}{m} \sum_{i=1}^{m} \ell_k(h(x_i), y_i). \tag{3}$$

We will show in Section 3 that, with high probability, $\mathcal{L}_\Lambda(h)$ is bounded by $\widehat{\mathcal{L}}_\Lambda(h)$ and other terms depending on the Rademacher complexity of the underlying function class.

Assume that the hypothesis set $\mathcal{H}$ is parameterized by a vector $w \in \mathcal{W}$, thus $h \in \mathcal{H}$ can be denoted as $h_w$. The (unregularized) optimization problem of the learner can formulated as follows:

$$\min_{w \in \mathcal{W}} \max_{\lambda \in \Lambda} \sum_{k=1}^{p} \lambda_k \underbrace{\frac{1}{m} \sum_{i=1}^{m} \ell_k(h_w(x_i), y_i)}_{\widehat{\mathcal{L}}_k(h_w)}. \tag{4}$$

Given a number of standard assumptions on $\mathcal{W}, \Lambda$ and the base loss functions, the optimization problem above is convex, and can be solved using gradient-based algorithms, as we show in Section 4.

## 2.1 Relationship with the Pareto-Optimal Frontier

In this section, we discuss the connection between the ALMO loss function and the Pareto frontier. In particular, we will show that the solution of the ALMO loss function is a point on the Pareto-optimal frontier.

The Pareto-optimal frontier is the set of feasible loss tuples $(\ell_1(h), \ldots, \ell_p(h))$, $h \in \mathcal{H}$, that are not *strictly dominated* by some other feasible tuple. The function $h \in \mathcal{H}$, or the corresponding tuple $(\ell_1(h), \ldots, \ell_p(h))$, is strictly dominated by $h' \in \mathcal{H}$, if $\ell_k(h') \leq \ell_k(h)$ for all $k \in [p]$ with $\ell_j(h') < \ell_j(h)$ for at least one index $j \in [p]$.

The following proves a key property of our agnostic solution, when it is unique. Note that the solution is unique, for example, for strictly convex losses.

**Proposition 1.** *Assume that the agnostic solution is unique, then it is Pareto-optimal.*

*Proof.* Assume that $h^* \in \mathcal{H}$ is a minimizer of the ALMO objective for $\{\ell_1, \ldots, \ell_p\}$. That is, for all $h \in \mathcal{H}$,

$$\min_{h \in \mathcal{H}} \mathcal{L}_\Lambda(h) = \sum_{k=1}^{p} \lambda_k^* \ell_k(h^*) \leq \sum_{k=1}^{p} \lambda_k^* \ell_k(h),$$

for some $\lambda^* \in \Lambda$. Let $h'$ be a strictly dominating point for $h^*$. That is, $\forall k \in [p], \ell_k(h') \leq \ell_k(h^*)$, with $\ell_j(h') < \ell_j(h^*)$ for at least one $j \in [p]$. Then, if $\lambda_j^* = 0$ for any $j$ such that $\ell_j(h') < \ell_j(h^*)$, then, $\ell_l(h') = \ell_l(h^*)$ for other indices $l$ and $\mathcal{L}_\Lambda(h') = \mathcal{L}_\Lambda(h^*)$. Since the solution is unique, we then have $h' = h^*$.

Otherwise, we must have:

$$\sum_{k=1}^{p} \lambda_k^* \ell_k(h') < \sum_{k=1}^{p} \lambda_k^* \ell_k(h^*) = \min_{h \in \mathcal{H}} \mathcal{L}_\Lambda(h),$$

which, in view of the definition of minimality of $h^*$, implies that $h'$ is not in $\mathcal{H}$, that $(\ell_1(h'), \ldots, \ell_p(h'))$ is not a feasible point, and that $h^*$ is Pareto-optimal. $\square$

Note that the uniqueness assumption of the theorem can be relaxed. In particular, it is not hard to show, using the same proof, that it suffices that there is an optimal $\lambda^*$ with $\lambda_j^* \neq 0$ for all $j$ to guarantee that any agnostic solution is pareto-optimal.

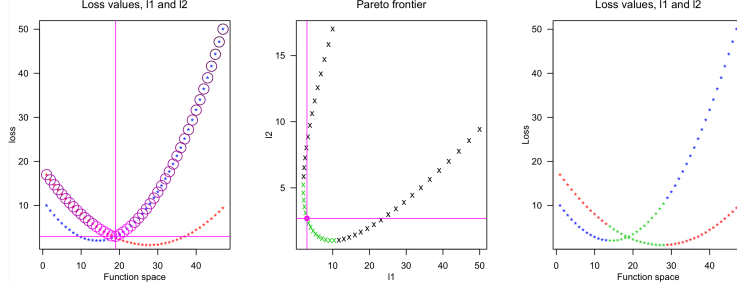

**Figure 1:** Illustration of the connection between the ALMO loss and the Pareto-optimal frontier. Left: two losses in a parameterized function space. The minimum of the ALMO loss is indicated by the purple lines; middle: the Pareto-optimal frontier. $\ell_1$ corresponds to the blue function in the left plot, $\ell_2$ corresponds to the red function; right: the optimal frontier indicated in the function space. See text for further explanation.

We illustrate the connection between the Pareto-optimal frontier and the optimizer of the ALMO loss with the plots in Figure 1 for two losses $\ell_1$ and $\ell_2$. The leftmost plot in Figure 1 illustrates the two losses, $\ell_1$ in blue and $\ell_2$ in red as a function of the parameterized function space. For a given function $h$ along the x-axis, the convex combination of the losses $\lambda\ell_1(h) + (1-\lambda)\ell_2(h)$ attains its maximum with respect to $\lambda$ at the value $\max\{\ell_1(h), \ell_2(h)\}$, circled in violet, with a value of $\lambda \in \{0,1\}$. The darker the violet color, the larger the value of the ALMO objective. The minimum over $h \in \mathcal{H}$ of the ALMO objective is attained for $\partial\mathcal{L}_\Lambda(h)/\partial\lambda = 0$, which is at the crossing point $\ell_1(h^*) = \ell_2(h^*)$ of the loss curves indicated by the crossing lines. At a crossing point, any value of $\lambda$ attains this minimum, a natural choice is to pick the $\lambda$ corresponding to uniform weights.

In the middle plot we illustrate the corresponding Pareto-optimal frontier of the points $(\ell_1, \ell_2)$ in green and indicate the ALMO solution. Finally, in the rightmost plot we tie back the Pareto-optimal frontier to the plot of the losses.

## 3 Learning Guarantees

In this section, we derive learning guarantees for the ALMO framework that rely on the Rademacher complexity of the family of loss functions and the mixture weights $\lambda$. The bounds that we show, being data-dependent, will motivate a scalable algorithm for the minimization of the agnostic multi-objective loss. The algorithm is presented in Section 4.

Let $\mathcal{G}$ denote the family of the losses associated to a hypothesis set $\mathcal{H}$: $\{\mathcal{G} : (x,y) \mapsto \ell(h(x), y) : \forall h \in \mathcal{H}\}$. Typically, generalization analysis in similar cases involves bounding the sample Rademacher complexity of $\mathcal{G}$, that is

$$\widehat{\mathfrak{R}}_S(\mathcal{G}) = \mathbb{E}_{\boldsymbol{\sigma}}\left[\sup_{h \in \mathcal{H}} \frac{1}{m}\sum_{i=1}^{m}\sigma_i\ell(h(x_i), y_i)\right], \tag{5}$$

where $\sigma_i$-s, $i \in [m]$, are independent uniformly distributed random variables taking values in $\{-1, +1\}$.

However, the agnostic multi-objective loss $\mathcal{L}_\Lambda(h)$ contains a maximum over $\lambda \in \Lambda$, which causes additional complication. For the proper analysis, we need to extend the definition of the sample Rademacher complexity by including the $\lambda_k$ terms as follows:

$$\widehat{\mathfrak{R}}_S(\mathcal{G}, \lambda) = \mathbb{E}_{\boldsymbol{\sigma}}\left[\sup_{h \in \mathcal{H}} \sum_{k=1}^{p}\lambda_k\frac{1}{m}\sum_{i=1}^{m}\sigma_i\ell_k(h(x_i), y_i)\right]. \tag{6}$$

By using $\widehat{\mathfrak{R}}_S(\mathcal{G}, \lambda)$ in the theorem below, we show the connection between the theoretical and empirical agnostic multi-objective losses $\mathcal{L}_\Lambda(h)$ and $\widehat{\mathcal{L}}_\Lambda(h)$. The theorem below is based on common steps in Rademacher complexity bound proofs [Koltchinskii and Panchenko, 2002, Hoffman et al., 2018, Mohri et al., 2012]. Additionally, for the theorem below, we assume that $\forall(x,y), (x',y') \in \mathcal{X} \times \mathcal{Y}, \forall h \in \mathcal{H} : ||(h(x'), y') - (h(x), y)|| \leq \mathcal{D}_\mathcal{H}$. This holds for all bounded hypothesis classes. For example, for the case of binary classification $\mathcal{D}_\mathcal{H} = O(\sqrt{C})$, where $C$ is the number of classes.

**Theorem 2.** *If the loss functions $\ell_k$ are $M_k$-Lipschitz and bounded by $M$, then for any $\epsilon > 0$ and $\delta > 0$, with probability at least $1 - \delta$, the following inequality holds for $\forall h \in \mathcal{H}$ and $\forall \lambda \in \Lambda$:*

$$\mathcal{L}_\lambda(h) \leq \widehat{\mathcal{L}}_\lambda(h) + 2\widehat{\mathfrak{R}}_S(\mathcal{G}, \lambda) + M\epsilon + \mathcal{D}_\mathcal{H} \sum_{k=1}^{p} \lambda_k M_k \sqrt{\frac{1}{2m} \log\left[\frac{|\Lambda|_\epsilon}{\delta}\right]}, \tag{7}$$

*where $\Lambda_\epsilon$ is an minimum $\epsilon$-cover of $\Lambda$.*

The proof of Theorem 2 is given in Appendix B.1 The bound in Theorem 2 is a function of $\lambda$. To make it uniform with respect to $\lambda$, we first observe that when $\ell_k$ are $M_k$-Lipschitz, by a direct application of Talagrand's contraction Lemma [Ledoux and Talagrand, 1991], the following holds:

$$\widehat{\mathfrak{R}}_S(\mathcal{G}, \lambda) \leq \sum_{k=1}^{p} \lambda_k M_k \widehat{\mathfrak{R}}_S(\mathcal{H}). \tag{8}$$

Thus, the only terms that are $\lambda$-dependent in the bound are of the form $\sum_{k=1}^{p} \lambda_k M_k$. This naturally leads us to control the complexity by imposing a regularization of the form $\sum_{k=1}^{p} \lambda_k M_k \leq \beta$, which leads to the following generalization bound for the agnostic multi-objective loss.

**Theorem 3.** *Let the loss functions $\ell_k$ be $M_k$-Lipschitz and bounded by $M$. Assume $\forall \lambda \in \Lambda : \sum_{k=1}^{p} \lambda_k M_k \leq \beta$, then for any $\epsilon > 0$ and $\delta > 0$, with probability at least $1 - \delta$, the following inequality holds for $\forall h \in \mathcal{H}$:*

$$\mathcal{L}_\Lambda(h) \leq \widehat{\mathcal{L}}_\Lambda(h) + 2\beta\widehat{\mathfrak{R}}_S(\mathcal{H}) + M\epsilon + \beta\mathcal{D}_\mathcal{H}\sqrt{\frac{1}{2m} \log\left[\frac{|\Lambda|_\epsilon}{\delta}\right]}. \tag{9}$$

The bound of Theorem 3 shows that, with high probability, the agnostic multi-objective loss $\mathcal{L}_\Lambda(h)$ is bounded by the empirical agnostic multi-objective loss $\widehat{\mathcal{L}}_\Lambda(h)$ and the sample-dependent complexity terms. The bound suggests that the learner in the ALMO framework should seek to find a hypothesis $h \in \mathcal{H}$ that provides the best trade-off between the empirical loss $\widehat{\mathcal{L}}_\lambda(h)$ and the Rademacher complexity. This means that the learner seeking to minimize $\mathcal{L}_\lambda(h)$ should minimize its empirical estimate $\widehat{\mathcal{L}}_\lambda(h)$ and apply a regularization to control the complexity terms. The bound suggests that the regularization should connect the properties of the constraint set $\Lambda$ and the Lipschitz properties of each individual base loss function in the form $\sum_{k=1}^{p} \lambda_k M_k \leq \beta$. This means that the ALMO algorithm should control the $M_k$-weighted norm of the mixture weights vector $\lambda$. Note that, for the cases where it is nontrivial to estimate the individual Lipschitz constants $M_k$, our theory allows us to use a simplified regularization $\|\lambda\|_q \leq \beta$ for some $q$-norm $\|\cdot\|_q$, which will also control $\sum_{k=1}^{p} \lambda_k M_k$ for bounded loss functions.

## 4 Algorithm

In this section, we describe the algorithm for the ALMO framework and the regularization inspired by the learning guarantees derived in Section 3. Moreover, we derive the convergence guarantees for the algorithm, which demonstrate that it scales well to large problems. To emphasize that for the optimization purpose $w \in \mathcal{W}$ and $\lambda \in \Lambda$ are real-valued vectors, we will use bold symbols for the pair $(\mathbf{w}, \boldsymbol{\lambda})$.

### 4.1 Regularization

The generalization bounds of Section 3 suggest minimizing the agnostic empirical loss $\widehat{\mathcal{L}}_\Lambda(h)$, while controlling for the complexity of hypothesis class $\mathcal{H}$. This naturally translates into the following regularized minimization problem in the ALMO framework:

$$\min_{h \in \mathcal{H}} \max_{\boldsymbol{\lambda} \in \text{conv}(\Lambda)} \widehat{\mathcal{L}}_\lambda(h) + \beta_1 \|h\|_\mathcal{H} + \beta_2 \sum_{k=1}^{p} \lambda_k M_k, \tag{10}$$

where $\beta_1, \beta_2 \geq 0$ are regularization parameters and $\|\cdot\|_\mathcal{H}$ is a norm defined on the hypothesis space $\mathcal{H}$. Note that, since $\widehat{\mathcal{L}}_\lambda(h)$ is linear in $\lambda$, we have replaced $\max_{\boldsymbol{\lambda} \in \Lambda}$ by $\max_{\boldsymbol{\lambda} \in \text{conv}(\Lambda)}$.

**Algorithm 1** ALMO. Initialize: $\mathbf{w}_0 \in \mathbf{W}, \boldsymbol{\lambda}_0 \in \Lambda$. Set step size $\gamma_{\boldsymbol{\lambda}}, \gamma_{\mathbf{w}}$.

---

**for** $t \in [1, T]$ **do**

$\quad \mathbf{w}_t \leftarrow \Pi_{\mathcal{W}}\left[w_{t-1} - \gamma_{\mathbf{w}} \delta_{\mathbf{w}} L(\mathbf{w}_{t-1}, \boldsymbol{\lambda}_{t-1})\right]$

$\quad \boldsymbol{\lambda}_t \leftarrow \Pi_{\Lambda}\left[\lambda_{t-1} - \gamma_{\boldsymbol{\lambda}} \delta_{\boldsymbol{\lambda}} L(\mathbf{w}_{t-1}, \boldsymbol{\lambda}_{t-1})\right]$

**end for**

$\mathbf{w}_T \leftarrow \frac{1}{T} \sum_{t=1}^{T} \mathbf{w}_t$

$\boldsymbol{\lambda}_T \leftarrow \frac{1}{T} \sum_{t=1}^{T} \boldsymbol{\lambda}_t$

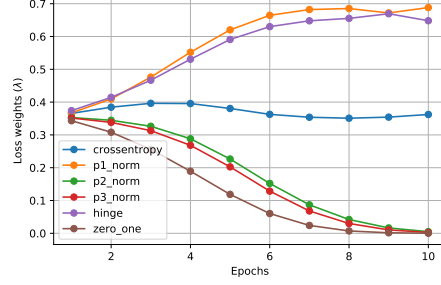

**Figure 2:** DNN training dynamics of mixture weights $\lambda_k$ on MNIST. Weights are logged at the end of each epoch.

Given a number of commonly used assumptions, the regularized problem in Equation 10 is convex. Assuming that the base loss functions $\ell_1, \ldots, \ell_p$ are convex in the first argument ensures that $\widehat{\mathcal{L}}_{\lambda}(h)$ is convex as well. Since the norm $\|h\|_{\mathcal{H}}$ is convex, then $\widehat{\mathcal{L}}_{\lambda}(h) + \beta_1 \|h\|_{\mathcal{H}}$ is a convex function of $h$. The maximum over $\boldsymbol{\lambda}$ of convex functions is a convex function itself. Thus, for a convex hypothesis space $\mathcal{H}$ the problem in Equation 10 is convex.

## 4.2 Optimization

The optimization problem of Equation 10 can be solved using projected gradient descent or more generally any mirror descent algorithm [Nemirovsky and Yudin, 1983]. For large-scale problems, we suggest to use the Stochastic Mirror-Prox gradient descent algorithm [Juditsky et al., 2011]. For the specific optimization problem presented in this paper, the Mirror-Prox algorithm admits a simplified form and can be implemented in common machine learning software that relies on symbolic gradient computation.

Assuming that the hypothesis set $\mathcal{H}$ is parameterized by a vector $\mathbf{w} \in \mathcal{W}$, we can simplify the notation to emphasize that the optimization is performed over $(\mathbf{w}, \boldsymbol{\lambda})$. Let $L(\mathbf{w}) = \frac{1}{m} \sum_{i=1}^{m} \ell(h_{\mathbf{w}}(x_i), y_i)$, then $L(\mathbf{w}, \boldsymbol{\lambda}) = \sum_{k=1}^{p} \lambda_k L_k(\mathbf{w})$. It is sufficient to present the algorithm for the unregularized problem

$$\min_{\mathbf{w} \in \mathcal{W}} \max_{\boldsymbol{\lambda} \in \Lambda} L(\mathbf{w}, \boldsymbol{\lambda}), \tag{11}$$

which can be extended in a straightforward manner to include the regularization term for $\mathbf{w}$ and $\boldsymbol{\lambda}$. Specifically, when a convex regularization term is added to the loss, without changing the constraint sets $\mathcal{W}$ and $\Lambda$, the problem is still convex.

Note that the problem above admits a game-theoretic interpretation. This is a two-player adversarial game, in which the first player chooses $\lambda \in \Lambda$ to maximize the objective in Equation 11, while the second player chooses $\mathbf{w} \in \mathcal{W}$ to minimize the loss. Thus, solving the optimization problem above provides the equilibrium of the two-player game.

Let $\nabla_{\mathbf{w}} L(\mathbf{w}, \boldsymbol{\lambda})$ be the gradient of the loss function $L(\mathbf{w}, \boldsymbol{\lambda})$ with respect to $\mathbf{w}$ and $\nabla_{\boldsymbol{\lambda}} L(\mathbf{w}, \boldsymbol{\lambda})$ be the gradient with respect to $\boldsymbol{\lambda}$. Let $\delta_{\mathbf{w}} L(\mathbf{w}, \boldsymbol{\lambda})$ and $\delta_{\boldsymbol{\lambda}} L(\mathbf{w}, \boldsymbol{\lambda})$ be the unbiased stochastic estimates of these gradients with respect to $\mathbf{w}$ and $\boldsymbol{\lambda}$. The Mirror-Prox descent adapted to the learning with multiple objectives problem obtains stochastic gradient estimates $\delta_{\mathbf{w}} L(\mathbf{w}, \boldsymbol{\lambda})$ and $\delta_{\boldsymbol{\lambda}} L(\mathbf{w}, \boldsymbol{\lambda})$ at each step $t \in [1, T]$ and then updates $(\mathbf{w}_t, \boldsymbol{\lambda}_t)$. After each update, the projections $\Pi_{\mathcal{W}}$ and $\Pi_{\Lambda}$ onto $\mathcal{W}$ and $\Lambda$ respectively are computed. If $\Lambda = \Delta_p$, then we can efficiently compute these projections [Duchi et al., 2008]. The pseudocode for the ALMO optimization algorithm is given in Algorithm 1.

## 4.3 Convergence Guarantees

We can show favorable convergence guarantees for the ALMO algorithm using a number of common assumptions, including the convexity of $L(\mathbf{w}, \boldsymbol{\lambda})$ in the first argument.

**Theorem 4.** *[See Appendix B.2 for the proof.] For $(\mathbf{w}, \boldsymbol{\lambda}) \mapsto L(\mathbf{w}, \boldsymbol{\lambda})$ convex in its first argument, assume the following: $\forall \mathbf{w} \in \mathcal{W}, \forall \boldsymbol{\lambda} \in \Lambda \colon \|\mathbf{w}\|_2 \leq \mathcal{D}_{\mathcal{W}}, \|\boldsymbol{\lambda}\|_2 \leq \mathcal{D}_{\Lambda}, \|\nabla_{\mathbf{w}} L(\mathbf{w}, \boldsymbol{\lambda})\|_2 \leq \mathcal{G}_{\mathbf{w}}, \|\nabla_{\boldsymbol{\lambda}} L(\mathbf{w}, \boldsymbol{\lambda})\|_2 \leq \mathcal{G}_{\boldsymbol{\lambda}}$. Let the variance of unbiased stochastic gradients be bounded by $\sigma_{\mathbf{w}}^2$ and $\sigma_{\boldsymbol{\lambda}}^2$ respectively. Then, for step sizes $\gamma_{\mathbf{w}} = \frac{1}{\sqrt{T}} \frac{2\mathcal{D}_{\mathcal{W}}}{\sqrt{\sigma_{\mathbf{w}}^2 + \mathcal{G}_{\mathbf{w}}^2}}$ and $\gamma_{\boldsymbol{\lambda}} = \frac{1}{\sqrt{T}} \frac{2\mathcal{D}_{\Lambda}}{\sqrt{\sigma_{\boldsymbol{\lambda}}^2 + \mathcal{G}_{\boldsymbol{\lambda}}^2}}$, the following convergence guarantees hold:*

$$\mathbb{E}\left[ \max_{\boldsymbol{\lambda} \in \Lambda} L(\mathbf{w}_T, \boldsymbol{\lambda}) - \min_{\mathbf{w} \in \mathcal{W}} \max_{\boldsymbol{\lambda} \in \Lambda} L(\mathbf{w}, \boldsymbol{\lambda}) \right] \leq \frac{1}{\sqrt{T}} \left( 3\mathcal{D}_{\mathcal{W}} \sqrt{\sigma_{\mathbf{w}}^2 + \mathcal{G}_{\mathbf{w}}^2} + 3\mathcal{D}_{\Lambda} \sqrt{\sigma_{\boldsymbol{\lambda}}^2 + \mathcal{G}_{\boldsymbol{\lambda}}^2} \right). \quad (12)$$

## 5 Experiments

The experiments presented in this section illustrate the benefits of the ALMO framework combined with the generalization guarantees derived in Section 3 and the stochastic algorithm in Section 4. Our experiments serve to illustrate the application of our novel agnostic learning formulation and to support the claim that our algorithm provides more robust results than training with a fixed mixture of base objectives. We present results for convex models and deep neural networks to demonstrate that our algorithm performs well even for non-convex hypothesis classes. The ALMO optimization algorithm is implemented in TENSORFLOW [Abadi et al., 2016] and KERAS [Chollet et al., 2015].

For the base losses $\ell_1, \ldots, \ell_p$, we use a variety of common training loss functions, each having its own advantages and drawbacks. For example, the Zero-One loss function often results in robust models when the data contains outliers, but it underperforms when the data is concentrated near decision surface [Zhao et al., 2010]. The opposite usually applies to the hinge loss function. To solve a complex machine learning problem, the learner would wish to combine multiple loss functions, making use of their strengths and mitigating weaknesses. For the experiments, each base objective $\ell_k$ (see Appendix, Table 3) is normalized so that $M_k \leq 1$ and hence $M = 1$.

For every dataset, we train a model by minimizing the empirical agnostic multi-objective loss $\widehat{\mathcal{L}}_h$ objective and we benchmark it against $p$ models trained with each individual $k$-th base loss function for $k \in [1, p]$. We also include a model trained with fixed uniform mixture weights $\lambda_k = 1/p$. Note, we do not include techniques based on searching the Pareto-efficient solutions for the mixture of losses, since the end goal of these frameworks is distinct from that of ALMO, in particular, a specific point still needs to be subsequently selected on the Pareto curve. For the connection between the two frameworks, see the discussion in Section 2.

The models are compared on MNIST [LeCun and Cortes, 2010], Fashion-MNIST [Xiao et al., 2017] and ADULT [Dua and Graff, 2017] datasets with standard feature preprocessing techniques applied. We report results for two model architectures: a logistic regression and a neural network with dimensions 1024-512-128. For both models, we run hyper-parameter tuning with a parameter grid size 50 on a validation set, which is 20% of the training data.

We report the average values of the cross-entropy, zero-one, hinge and 1-norm losses in Tables 1 and 2. We also report the AUC (area under the ROC curve), which was not included in the training losses. The standard deviations are obtained by retraining every model 3 times.

Experimental results support our claim that the ALMO framework improves the model for the worst performing loss. The hinge loss is often observed with a significant loss value and, as illustrated in Figure 2, the corresponding $\lambda$ attains a high value. In the appendix, we provide results from training on just one loss at a time and discuss the improvements of ALMO as compared to that baseline.

The resulting model is robust and avoids selecting a specific set of mixture weights for training. It also performs better than the baseline in terms of AUC on all datasets, while AUC was not used for training. Additionally, the ALMO algorithm can be used as a tool for the selection of base objectives (e.g., AutoML), as ALMO during training increases the mixture weights for the worst performing base losses, while the others are decreasing (see Figure 2). As expected, losses trained with a higher mixture weight show an improvement compared to the uniform case. Even for cases like cross-entropy in the DNN-MNIST configuration (see Table 2), the algorithm balances the 1-norm loss, and AUC shows a more robust model. Note the significant improvement in AUC for the DNN-Adult dataset, which may be related to the dramatic decrease of the hinge loss. Finally, as shown in both tables, ALMO does not select to optimize the 1-norm loss, assigns it a low mixture weight and does not improve over uniform.

**Table 1:** Comparison of loss functions for logistic regression model on the test set.

|  | MNIST | | Fashion MNIST | | Adult | |
|---|---|---|---|---|---|---|
|  | Uniform | ALMO | Uniform | ALMO | Uniform | ALMO |
| **Cross-entropy** | 0.1412 | **0.0726** | 0.1011 | **0.0716** | 0.680 | **0.501** |
| (std) | (0.0004) | (0.0005) | (0.0002) | (0.0002) | (0.002) | (0.008) |
| **Zero-one** | 0.143 | **0.114** | 0.1836 | **0.1789** | 0.224 | **0.214** |
| (std) | (0.003) | (0.002) | (0.0005) | (0.0008) | (0.002) | (0.002) |
| **Hinge** | 0.839 | **0.711** | **0.633** | 0.695 | 0.967 | **0.518** |
| (std) | (0.003) | (0.005) | (0.009) | (0.011) | (0.004) | (0.021) |
| **1-norm** | **0.2349** | 0.6587 | **0.488** | 0.672 | **0.250** | 0.871 |
| (std) | (0.001) | (0.007) | (0.008) | (0.005) | (0.003) | (0.056) |
| **AUC** | 0.9801 | **0.9877** | 0.9776 | **0.9794** | 0.8291 | **0.8321** |
| (std) | (0.0002) | (0.0002) | (0.0007) | (0.0001) | (0.0018) | (0.0003) |

**Table 2:** Comparison of loss functions for DNN model on the test set.

|  | MNIST | | Fashion MNIST | | Adult | |
|---|---|---|---|---|---|---|
|  | Uniform | ALMO | Uniform | ALMO | Uniform | ALMO |
| **Cross-entropy** | **0.044** | 0.045 | 0.0700 | **0.0614** | 0.673 | **0.460** |
| (std) | (0.003) | (0.005) | (0.0005) | (0.0009) | (0.001) | (0.003) |
| **Zero-one** | 0.0158 | **0.0153** | 0.114 | **0.110** | **0.149** | 0.159 |
| (std) | (0.0004) | (0.0002) | (0.003) | (0.004) | (0.002) | (0.002) |
| **Hinge** | **0.631** | 0.752 | 0.661 | **0.547** | 0.914 | **0.389** |
| (std) | (0.002) | (0.041) | (0.004) | (0.008) | (0.001) | (0.006) |
| **1-norm** | 0.436 | **0.313** | **0.435** | 0.554 | **0.153** | 0.845 |
| (std) | (0.002) | (0.038) | (0.006) | (0.0010) | (0.001) | (0.022) |
| **AUC** | 0.9986 | **0.9993** | 0.9883 | **0.9911** | 0.8092 | **0.8840** |
| (std) | (0.0002) | (0.0001) | (0.0002) | (0.0005) | (0.0004) | (0.0031) |

# 6 Conclusion

We introduced a new framework (ALMO) for multi-objective learning that is robust against any mixture distribution of the base objectives, hence avoiding the subjective step of selecting mixture coefficients for multi-loss training. We have given a detailed theoretical analysis and a learning algorithm motivated by the theory. The algorithm is based on stochastic gradient descent, therefore applicable to a wide range of large-scale domains and can be easily implemented in popular computational frameworks. The experiments show that the ALMO framework builds more robust models for a variety of objectives in different machine learning problems. The agnostic framework introduced here is directly applicable and beneficial not only to standard supervised learning settings, but also to other scenarios where a learner is seeking to combine multiple objectives, such as transfer learning and domain adaptation.

## Broader Impact

This paper presents a novel approach for learning with multiple losses. The algorithm is robust in the sense that it optimizes for the most adversarial mixture of the included losses. It furthermore demonstrates good performance on losses not included in the optimization. This is an important problem from a fairness perspective, where multiple losses are often at play and different interest groups may disagree on what loss to optimize for. An illustrative example is the analysis of the COMPAS tool for predicting recidivism by Angwin et al. [2019] demonstrating how different interest groups would have wanted to optimize for different losses. Not all losses can be optimized for simultaneously, as proven by Kleinberg et al. [2017], but to the extent that this is possible, our algorithm provides a step in the direction of guarding against the most unfavorable condition.

## Acknowledgments

We thank Ananda Theertha Suresh for discussions on related topics. The work of MM and DS was partly supported by NSF CCF-1535987, NSF IIS-1618662, and a Google Research Award.

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
