[Supplementary Material]

# A  Losses

Table 3 lists the losses used for training.

**Table 3:** Base loss functions used for experiments.

| Base loss function | $\ell(h(x), y)$ |
|---|---|
| Zero-One | $l_1 = \begin{cases} 0, & \text{if } yh(x) \geq 0 \\ 1, & \text{if } yh(x) < 0 \end{cases}$ |
| Hinge | $l_2 = \begin{cases} 0, & \text{if } yh(x) \geq 1 \\ \max(0, 1 - yh(x)), & \text{if } yh(x) > 1 \end{cases}$ |
| Norm $q$ | $l_3 = \|h(x) - y\|_q^q$ |
| Cross-Entropy | $l_5 = \log(1 + \exp(-yh(x)))$ |

# B  Proof of Theorems

## B.1  Proof of Theorem 2

If the loss functions $\ell_k$ are $M_k$-Lipschitz and bounded by $M$, then for any $\epsilon > 0$ and $\delta > 0$, with probability at least $1 - \delta$, the following inequality holds for $\forall h \in \mathcal{H}$ and $\forall \lambda \in \Lambda$:

$$\mathcal{L}_\lambda(h) \leq \widehat{\mathcal{L}}_\lambda(h) + 2\widehat{\mathfrak{R}}_S(\mathcal{G}, \lambda) + M\epsilon + \mathcal{D}_{\mathcal{H}} \sum_{k=1}^{p} \lambda_k M_k \sqrt{\frac{1}{2m} \log\left[\frac{|\Lambda|_\epsilon}{\delta}\right]}, \tag{13}$$

where $\Lambda_\epsilon$ is an minimum $\epsilon$-cover of $\Lambda$.

*Proof.* For any $\lambda \in \Lambda$ and sample $S = \{(x_1, y_1), \ldots, (x_m, y_m)\}$, let $\Psi(S) = \sup_{h \in \mathcal{H}} \mathcal{L}_\lambda(h) - \widehat{\mathcal{L}}_\lambda(h)$. Let $S'$ be a sample different from $S$ by only one point $(x', y')$, then

$$\begin{aligned}
\Psi(S') - \Psi(S) &= \sup_{h \in \mathcal{H}} \left[\mathcal{L}_\lambda(h) - \widehat{\mathcal{L}}'_\lambda(h)\right] - \sup_{h \in \mathcal{H}} \left[\mathcal{L}_\lambda(h) - \widehat{\mathcal{L}}_\lambda(h)\right] \\
&\leq \sup_{h \in \mathcal{H}} \left[\mathcal{L}_\lambda(h) - \widehat{\mathcal{L}}'_\lambda(h) - \mathcal{L}_\lambda(h) + \widehat{\mathcal{L}}_\lambda(h)\right] \\
&= \sup_{h \in \mathcal{H}} \left[\widehat{\mathcal{L}}_\lambda(h) - \widehat{\mathcal{L}}'_\lambda(h)\right] \\
&= \sup_{h \in \mathcal{H}} \left[\sum_{k=1}^{p} \lambda_k \frac{1}{m} \sum_{i=1}^{m} \ell_k(h(x'_i), y'_i) - \sum_{k=1}^{p} \lambda_k \frac{1}{m} \sum_{i=1}^{m} \ell_k(h(x_i), y_i)\right] \\
&= \frac{1}{m} \sup_{h \in \mathcal{H}} \sum_{k=1}^{p} \lambda_k \left(\ell_k(h(x'_i, y'_i)) - \ell_k(h(x_i), y_i)\right) \\
&\leq \frac{1}{m} \sup_{h \in \mathcal{H}} \sum_{k=1}^{p} \lambda_k M_k \|[h(x'_i), y'_i] - [h(x_i), y_i]\| \\
&\leq \frac{1}{m} \sup_{h \in \mathcal{H}} \sum_{k=1}^{p} \lambda_k M_k \mathcal{D}_{\mathcal{H}} \\
&\leq \frac{\mathcal{D}_{\mathcal{H}}}{m} \sum_{k=1}^{p} \lambda_k M_k.
\end{aligned}$$

By McDiarmid's inequality, for any $\delta > 0$ with probability at least $1 - \delta$ for any $h \in \mathcal{H}$:

$$\mathcal{L}_\lambda(h) \leq \widehat{\mathcal{L}}_\lambda(h) + \mathbb{E}\left[\sup_{h \in \mathcal{H}} \mathcal{L}_\lambda(h) - \widehat{\mathcal{L}}_\lambda(h)\right] + \mathcal{D}_{\mathcal{H}} \sum_{k=1}^{p} \lambda_k M_k \sqrt{\frac{1}{2m} \log \frac{1}{\delta}}.$$

The inequality above holds for a particular value of $\lambda$ fixed in advance. Using the union over the minimum $\epsilon$-cover $\Lambda_\epsilon$, with probability at least $1 - \delta$ for any $\lambda \in \Lambda_\epsilon$ and $h \in \mathcal{H}$:

$$\mathcal{L}_\lambda(h) \leq \widehat{\mathcal{L}}_\lambda(h) + \mathbb{E}\left[\sup_{h \in \mathcal{H}} \mathcal{L}_\lambda(h) - \widehat{\mathcal{L}}_\lambda(h)\right] + \mathcal{D}_\mathcal{H} \sum_{k=1}^{p} \lambda_k M_k \sqrt{\frac{1}{2m} \log \frac{|\Lambda_\epsilon|}{\delta}}.$$

Using the definition of the minimum $\epsilon$-cover, and also recalling that $\mathbb{E}\left[\sup_{h \in \mathcal{H}} \mathcal{L}_\lambda(h) - \widehat{\mathcal{L}}_\lambda(h)\right] \leq \widehat{\mathfrak{R}}_S(\mathcal{G}, \lambda)$, with probability at least $1 - \delta$ for any $\lambda \in \Lambda$ and $h \in \mathcal{H}$:

$$\mathcal{L}_\lambda(h) \leq \widehat{\mathcal{L}}_\lambda(h) + \mathbb{E}\left[\sup_{h \in \mathcal{H}} \mathcal{L}_\lambda(h) - \widehat{\mathcal{L}}_\lambda(h)\right] + M\epsilon + \mathcal{D}_\mathcal{H} \sum_{k=1}^{p} \lambda_k M_k \sqrt{\frac{1}{2m} \log \frac{|\Lambda_\epsilon|}{\delta}}$$

$$\leq \widehat{\mathfrak{R}}_S(\mathcal{G}, \lambda) + M\epsilon + \mathcal{D}_\mathcal{H} \sum_{k=1}^{p} \lambda_k M_k \sqrt{\frac{1}{2m} \log \frac{|\Lambda_\epsilon|}{\delta}}.$$

$\square$

## B.2 Proof of Theorem 4

For $L(\mathbf{w}, \boldsymbol{\lambda})$ convex in the first argument, assume $\forall \mathbf{w} \in \mathcal{W}, \forall \boldsymbol{\lambda} \in \Lambda : \|\mathbf{w}\|_2 \leq \mathcal{D}_\mathcal{W}, \|\boldsymbol{\lambda}\|_2 \leq \mathcal{D}_\Lambda, \|\nabla_\mathbf{w} L(\mathbf{w}, \boldsymbol{\lambda})\|_2 \leq \mathcal{G}_\mathbf{w}, \|\nabla_\boldsymbol{\lambda} L(\mathbf{w}, \boldsymbol{\lambda})\|_2 \leq \mathcal{G}_\boldsymbol{\lambda}$. Let the variance of unbiased stochastic gradients be bounded by $\sigma_\mathbf{w}^2$ and $\sigma_\boldsymbol{\lambda}^2$ respectively. If the step sizes are $\gamma_\mathbf{w} = \frac{1}{\sqrt{T}} \frac{2\mathcal{D}_\mathcal{W}}{\sqrt{\sigma_\mathbf{w}^2 + \mathcal{G}_\mathbf{w}^2}}$ and $\gamma_\boldsymbol{\lambda} = \frac{1}{\sqrt{T}} \frac{2\mathcal{D}_\Lambda}{\sqrt{\sigma_\boldsymbol{\lambda}^2 + \mathcal{G}_\boldsymbol{\lambda}^2}}$, then the following convergence guarantees apply for the ALMO algorithm:

$$\mathbb{E}\left[\max_{\boldsymbol{\lambda} \in \Lambda} L(\mathbf{w}_T, \lambda) - \min_{\mathbf{w} \in \mathcal{W}} \max_{\boldsymbol{\lambda} \in \Lambda} L(\mathbf{w}, \boldsymbol{\lambda})\right] \leq \frac{1}{\sqrt{T}}\left(3\mathcal{D}_\mathcal{W}\sqrt{\sigma_\mathbf{w}^2 + \mathcal{G}_\mathbf{w}^2} + 3\mathcal{D}_\Lambda\sqrt{\sigma_\boldsymbol{\lambda}^2 + \mathcal{G}_\boldsymbol{\lambda}^2}\right). \quad (14)$$

*Proof.*

$$\begin{aligned}
\max_{\boldsymbol{\lambda} \in \Lambda} L(\mathbf{w}_T, \lambda) - \min_{\mathbf{w} \in \mathcal{W}} \max_{\boldsymbol{\lambda} \in \Lambda} L(\mathbf{w}, \boldsymbol{\lambda}) &= \max_{\boldsymbol{\lambda} \in \Lambda} L(\mathbf{w}_T, \lambda) - \max_{\boldsymbol{\lambda} \in \Lambda} \min_{\mathbf{w} \in \mathcal{W}} L(\mathbf{w}, \boldsymbol{\lambda}) \\
&\leq \max_{\boldsymbol{\lambda} \in \Lambda}\left[L(\mathbf{w}_T, \lambda) - \min_{\mathbf{w} \in \mathcal{W}} L(\mathbf{w}, \boldsymbol{\lambda}_T)\right] \\
&= \max_{\boldsymbol{\lambda} \in \Lambda, \mathbf{w} \in \mathcal{W}}\left[L(\mathbf{w}_T, \lambda) - L(\mathbf{w}, \boldsymbol{\lambda}_T)\right] \\
&\leq \frac{1}{T} \max_{\boldsymbol{\lambda} \in \Lambda, \mathbf{w} \in \mathcal{W}}\left[\sum_{t=1}^{T} L(\mathbf{w}_t, \lambda) - L(\mathbf{w}, \boldsymbol{\lambda}_t)\right]
\end{aligned}$$

The last inequality follows from the convexity in $\mathbf{w}$. Given the resulting inequality above, the next step is to bound the difference $L(\mathbf{w}_t, \lambda) - L(\mathbf{w}, \boldsymbol{\lambda}_t)$ for each $t \in [1, T]$, using the standard techniques in convex optimization proofs.

$$\begin{aligned}
L(\mathbf{w}_t, \boldsymbol{\lambda}) - L(\mathbf{w}, \boldsymbol{\lambda}_t) &= L(\mathbf{w}_t, \boldsymbol{\lambda}) - L(\mathbf{w}_t, \boldsymbol{\lambda}_t) + L(\mathbf{w}_t, \boldsymbol{\lambda}_t) - L(\mathbf{w}, \boldsymbol{\lambda}_t) \\
&\leq (\boldsymbol{\lambda} - \boldsymbol{\lambda}_t)\nabla_\boldsymbol{\lambda} L(\mathbf{w}_t, \boldsymbol{\lambda}_t) + (\mathbf{w}_t - \mathbf{w})\nabla_\mathbf{w} L(\mathbf{w}_t, \boldsymbol{\lambda}_t) \\
&\leq (\boldsymbol{\lambda} - \boldsymbol{\lambda}_t)\delta_\boldsymbol{\lambda} L(\mathbf{w}_t, \boldsymbol{\lambda}_t) + (\mathbf{w}_t - \mathbf{w})\delta_\mathbf{w} L(\mathbf{w}_t, \boldsymbol{\lambda}_t) \\
&\quad + (\boldsymbol{\lambda} - \boldsymbol{\lambda}_t)(\nabla_\boldsymbol{\lambda} L(\mathbf{w}_t, \boldsymbol{\lambda}_t) - \delta_\boldsymbol{\lambda} L(\mathbf{w}_t, \boldsymbol{\lambda}_t)) \\
&\quad + (\mathbf{w}_t - \mathbf{w})(\nabla_\mathbf{w} L(\mathbf{w}_t, \boldsymbol{\lambda}_t) - \delta_\mathbf{w} L(\mathbf{w}_t, \boldsymbol{\lambda}_t))
\end{aligned}$$

Given the bound on $L(\mathbf{w}_t, \boldsymbol{\lambda}) - L(\mathbf{w}, \boldsymbol{\lambda}_t)$ we can obtain the following series of inequalities:

$$\max_{\boldsymbol{\lambda} \in \Lambda, \mathbf{w} \in \mathcal{W}} \left[ \sum_{t=1}^T L(\mathbf{w}_t, \boldsymbol{\lambda}) - L(\mathbf{w}, \boldsymbol{\lambda}_t) \right]$$

$$\leq \underbrace{\max_{\boldsymbol{\lambda} \in \Lambda, \mathbf{w} \in \mathcal{W}} \sum_{t=1}^T (\boldsymbol{\lambda} - \boldsymbol{\lambda}_t) \delta_{\boldsymbol{\lambda}} L(\mathbf{w}_t, \boldsymbol{\lambda}_t) + (\mathbf{w}_t - \mathbf{w}) \delta_{\mathbf{w}} L(\mathbf{w}_t, \boldsymbol{\lambda}_t)}_{A}$$

$$+ \underbrace{\max_{\boldsymbol{\lambda} \in \Lambda, \mathbf{w} \in \mathcal{W}} \sum_{t=1}^T \boldsymbol{\lambda} (\nabla_{\boldsymbol{\lambda}} L(\mathbf{w}_t, \boldsymbol{\lambda}_t) - \delta_{\boldsymbol{\lambda}} L(\mathbf{w}_t, \boldsymbol{\lambda}_t)) - \mathbf{w} (\nabla_{\mathbf{w}} L(\mathbf{w}_t, \boldsymbol{\lambda}_t) - \delta_{\mathbf{w}} L(\mathbf{w}_t, \boldsymbol{\lambda}_t))}_{B}$$

$$+ \underbrace{\sum_{t=1}^T \boldsymbol{\lambda}_t (\nabla_{\boldsymbol{\lambda}} L(\mathbf{w}_t, \boldsymbol{\lambda}_t) - \delta_{\boldsymbol{\lambda}} L(\mathbf{w}_t, \boldsymbol{\lambda}_t)) - \mathbf{w}_t (\nabla_{\mathbf{w}} L(\mathbf{w}_t, \boldsymbol{\lambda}_t) - \delta_{\mathbf{w}} L(\mathbf{w}_t, \boldsymbol{\lambda}_t))}_{C}$$

To complete the proof, we need to bound each of the terms $A, B, C$ in the sum above and take expectation. First, we show the bounds on $A$ as follows:

$$\mathbb{E} \left[ \max_{\boldsymbol{\lambda} \in \Lambda, \mathbf{w} \in \mathcal{W}} \sum_{t=1}^T (\boldsymbol{\lambda} - \boldsymbol{\lambda}_t) \delta_{\boldsymbol{\lambda}} L(\mathbf{w}_t, \boldsymbol{\lambda}_t) \right]$$

and

$$\mathbb{E} \left[ \max_{\boldsymbol{\lambda} \in \Lambda, \mathbf{w} \in \mathcal{W}} \sum_{t=1}^T (\mathbf{w}_t - \mathbf{w}) \delta_{\mathbf{w}} L(\mathbf{w}_t, \boldsymbol{\lambda}_t) \right],$$

which can be obtained in a similar way. Consider the following series of inequalities:

$$(\mathbf{w}_t - \mathbf{w}) \delta_{\mathbf{w}} L(\mathbf{w}_t, \boldsymbol{\lambda}_t)$$

$$= \frac{1}{2\gamma_{\mathbf{w}}} \sum_{t=1}^T \|\mathbf{w} - \mathbf{w}_t\|_2^2 + \gamma_{\mathbf{w}}^2 \|\delta_{\mathbf{w}} L(\mathbf{w}_t, \boldsymbol{\lambda}_t)\|_2^2 - \|\mathbf{w}_t - \gamma_{\mathbf{w}} \delta_{\mathbf{w}} L(\mathbf{w}_t, \boldsymbol{\lambda}_t) - \mathbf{w}\|_2^2$$

$$\leq \frac{1}{2\gamma_{\mathbf{w}}} \sum_{t=1}^T \|\mathbf{w} - \mathbf{w}_t\|_2^2 + \gamma_{\mathbf{w}}^2 \|\delta_{\mathbf{w}} L(\mathbf{w}_t, \boldsymbol{\lambda}_t)\|_2^2 - \|\mathbf{w}_{t+1} - \mathbf{w}\|_2^2$$

$$= \frac{1}{2\gamma_{\mathbf{w}}} \|\mathbf{w}_1 - \mathbf{w}\|_2^2 - \|\mathbf{w}_{T+1} - \mathbf{w}\|_2^2 + \frac{\gamma_{\mathbf{w}}}{2} \sum_{t=1}^T \|\delta_{\mathbf{w}} L(\mathbf{w}_t, \boldsymbol{\lambda}_t)\|_2^2$$

$$\leq \frac{1}{2\gamma_{\mathbf{w}}} \|\mathbf{w}_1 - \mathbf{w}\|_2^2 + \frac{\gamma_{\mathbf{w}}}{2} \sum_{t=1}^T \|\delta_{\mathbf{w}} L(\mathbf{w}_t, \boldsymbol{\lambda}_t)\|_2^2$$

$$\leq \frac{2R_{\mathcal{W}}}{\gamma_{\mathbf{w}}} + \frac{\gamma_{\mathbf{w}}}{2} \sum_{t=1}^T \|\delta_{\mathbf{w}} L(\mathbf{w}_t, \boldsymbol{\lambda}_t)\|_2^2$$

$$\leq \frac{2R_{\mathcal{W}}}{\gamma_{\mathbf{w}}} + \frac{\gamma_{\mathbf{w}}}{2} \sum_{t=1}^T \|\delta_{\mathbf{w}} L(\mathbf{w}_t, \boldsymbol{\lambda}_t) - \nabla_{\mathbf{w}} L(\mathbf{w}_t, \boldsymbol{\lambda}_t) + \nabla_{\mathbf{w}} L(\mathbf{w}_t, \boldsymbol{\lambda}_t)\|_2^2$$

Taking the maximum of both sides with respect to $\mathbf{w}$ and the expectation yields

$$\mathbb{E} \left[ \max_{\boldsymbol{\lambda} \in \Lambda, \mathbf{w} \in \mathcal{W}} \sum_{t=1}^T (\mathbf{w}_t - \mathbf{w}) \delta_{\mathbf{w}} L(\mathbf{w}_t, \boldsymbol{\lambda}_t) \right] \leq \frac{1}{2} \left( \frac{4\mathcal{D}_{\mathcal{W}}^2}{\gamma_{\mathbf{w}}} + \gamma_{\mathbf{w}} T \sigma_{\mathbf{w}}^2 + \gamma_{\mathbf{w}} T \mathcal{G}_{\mathbf{w}}^2 \right)$$

and a repeating the same steps for $\boldsymbol{\lambda}$ we obtain

$$\mathbb{E}\left[\max_{\boldsymbol{\lambda}\in\Lambda,\mathbf{w}\in\mathcal{W}}\sum_{t=1}^{T}(\boldsymbol{\lambda}-\boldsymbol{\lambda}_t)\delta_{\boldsymbol{\lambda}}L(\mathbf{w}_t,\boldsymbol{\lambda}_t)\right]\leq\frac{1}{2}\left(\frac{4\mathcal{D}_\Lambda^2}{\gamma_{\boldsymbol{\lambda}}}+\gamma_{\boldsymbol{\lambda}}T\sigma_{\boldsymbol{\lambda}}^2+\gamma_{\boldsymbol{\lambda}}T\mathcal{G}_{\boldsymbol{\lambda}}^2\right)$$

Next, we bound $B$ in the following way:

$$\max_{\boldsymbol{\lambda}\in\Lambda,\mathbf{w}\in\mathcal{W}}\sum_{t=1}^{T}\boldsymbol{\lambda}(\nabla_\lambda L(\mathbf{w}_t,\boldsymbol{\lambda}_t)-\delta_\lambda L(\mathbf{w}_t,\boldsymbol{\lambda}_t))\leq R_\Lambda\|\sum_{t=1}^{T}\boldsymbol{\lambda}(\nabla_\lambda L(\mathbf{w}_t,\boldsymbol{\lambda}_t)-\delta_\lambda L(\mathbf{w}_t,\boldsymbol{\lambda}_t)\|_2$$

After we take expectation of both sides, we get

$$\mathbb{E}\left[\max_{\boldsymbol{\lambda}\in\Lambda,\mathbf{w}\in\mathcal{W}}\sum_{t=1}^{T}\boldsymbol{\lambda}(\nabla_\lambda L(\mathbf{w}_t,\boldsymbol{\lambda}_t)-\delta_\lambda L(\mathbf{w}_t,\boldsymbol{\lambda}_t))\right]\leq\mathcal{D}_\Lambda\sqrt{T}\sigma_{\boldsymbol{\lambda}}$$

and in a completely similar way we can derive that

$$\mathbb{E}\left[\max_{\boldsymbol{\lambda}\in\Lambda,\mathbf{w}\in\mathcal{W}}\sum_{t=1}^{T}\mathbf{w}(\nabla_\mathbf{w} L(\mathbf{w}_t,\boldsymbol{\lambda}_t)-\delta_\mathbf{w} L(\mathbf{w}_t,\boldsymbol{\lambda}_t))\right]\leq\mathcal{D}_\mathcal{W}\sqrt{T}\sigma_{\mathbf{w}} \tag{15}$$

For the term $C$, it directly follows from the unbiased stochastic gradients that $\mathbb{E}[C]=0$. If we combine the bounds on $A,B,C$ that we derived above and let the step sizes be $\gamma_\mathbf{w}=\frac{1}{\sqrt{T}}\frac{2\mathcal{D}_\mathcal{W}}{\sqrt{\sigma_\mathbf{w}^2+\mathcal{G}_\mathbf{w}^2}}$ and $\gamma_{\boldsymbol{\lambda}}=\frac{1}{\sqrt{T}}\frac{2\mathcal{D}_\Lambda}{\sqrt{\sigma_{\boldsymbol{\lambda}}^2+\mathcal{G}_{\boldsymbol{\lambda}}^2}}$, we immediately obtain the final result. $\qquad\square$

**Table 4:** Comparison of logistic regression models trained with individual losses for the MNIST dataset.

| Model / Metric | Zero-one | Hinge | Cross-entropy | AUC |
|---|---|---|---|---|
| One Loss | **0.0747** | - | - | - |
| (std) | (0.0001) | - | - | - |
| Hinge | - | **0.0656** | - | - |
| (std) | - | (0.0002) | - | - |
| Uniform | 0.1187 | 0.5550 | 0.0911 | 0.9859 |
| (std) | (0.0020) | (0.0005) | (0.0005) | (0.0004) |
| ALMO | 0.1030 | 0.9228 | 0.0489 | 0.9905 |
| (std) | (0.0020) | (0.0050) | (0.0005) | (0.0002) |
| $\lambda$ | 0.0000 | 0.0478 | 0.8042 | - |

**Table 5:** Comparison of logistic regression models trained with individual losses for the Fashion-MNIST dataset.

| Model / Metric | Zero-one | Hinge | Cross-entropy | AUC |
|---|---|---|---|---|
| Zero-one | **0.1603** | - | - | - |
| (std) | (0.0005) | - | - | - |
| Hinge | - | **0.0958** | - | - |
| (std) | - | (0.0006) | - | - |
| Uniform | 0.1814 | 0.5431 | 0.0932 | 0.9786 |
| (std) | (0.0005) | (0.0035) | (0.0002) | (0.0003) |
| ALMO | 0.1774 | 0.4078 | 0.0683 | 0.9800 |
| (std) | (0.0008) | (0.0011) | (0.0002) | (0.0001) |
| $\lambda$ | 0.0000 | 0.1043 | 0.6410 | - |

# C  Additional Experiments

In this section, we present additional baseline studies to further highlight the benefits of the proposed ALMO algorithm. As baselines, we train with just one loss at a time and compare the ALMO performance to this per-loss optimal performance. This experimental setup is the same as the one detailed in the main section, but the realizations of the split of the data differs, which accounts for the small performance differences as compared to the tables in the main section.

In these tables, the boldfaced numbers indicate the performance of a classifier trained just for that loss. The ALMO algorithm often achieves a performance close to these values, without sacrificing any loss significantly. The mean values of the corresponding (un-normalized) $\lambda$-s are also reported to illustrate the weight ALMO assigns to the given loss. In a few cases, especially on the Adult dataset, the ALMO algorithm appears to be performing even slightly better than the baseline. We attribute this discrepancy to the non-convexity of the optimization problem and the small size of the Adult dataset.

**Table 6:** Comparison of logistic regression models trained with individual losses for the Adult dataset.

| Model / Metric | Zero-one | Hinge | Cross-entropy | AUC |
|---|---|---|---|---|
| Zero-one | **0.1849** | - | - | - |
| (std) | (0.0007) | - | - | - |
| Hinge | - | **0.4146** | - | - |
| (std) | - | (0.0001) | - | - |
| Uniform | 0.2001 | 0.8624 | 0.6320 | 0.8378 |
| (std) | (0.0008) | (0.0067) | (0.0033) | (0.0004) |
| ALMO | 0.1938 | 0.4085 | 0.4232 | 0.8409 |
| (std) | (0.0020) | (0.0021) | (0.0080) | (0.0003) |
| $\lambda$ | 0.0000 | 0.1317 | 0.0000 | - |

**Table 7:** Comparison of DNN models trained with individual losses for the MNIST dataset.

| Model / Metric | Zero-one | Hinge | Cross-entropy | AUC |
|---|---|---|---|---|
| Zero-one | **0.0215** | - | - | - |
| (std) | (0.0022) | - | - | - |
| Hinge | - | **0.0132** | - | - |
| (std) | - | (0.0002) | - | - |
| Uniform | 0.0168 | 0.6255 | 0.0437 | 0.9984 |
| (std) | (0.0013) | (0.0031) | (0.0008) | (0.0004) |
| ALMO | 0.0143 | 0.0092 | 0.0397 | 0.9996 |
| (std) | (0.0001) | (0.0001) | (0.0040) | (0.0001) |
| $\lambda$ | 0.0001 | 0.3893 | 0.1905 | - |

**Table 8:** Comparison of DNN models trained with individual losses for the Fashion-MNIST dataset.

| Model / Metric | Zero-one | Hinge | Cross-entropy | AUC |
|---|---|---|---|---|
| Zero-one | **0.1137** | - | - | - |
| (std) | (0.0010) | - | - | - |
| Hinge | - | **0.0595** | - | - |
| (std) | - | (0.0016) | - | - |
| Uniform | 0.1111 | 0.6603 | 0.0695 | 0.9889 |
| (std) | (0.0025) | (0.0030) | (0.0005) | (0.0006) |
| ALMO | 0.1077 | 0.1085 | 0.0374 | 0.9908 |
| (std) | (0.0035) | (0.0030) | (0.0008) | (0.0005) |
| $\lambda$ | 0.0000 | 0.1779 | 0.5199 | - |

**Table 9:** Comparison of DNN models trained with individual losses for the Adult dataset.

| Model / Metric | Zero-one | Hinge | Cross-entropy | AUC |
|---|---|---|---|---|
| Zero-one | **0.1564** | - | - | - |
| (std) | (0.0018) | - | - | - |
| Hinge | - | **0.5349** | - | - |
| (std) | - | (0.0176) | - | - |
| Uniform | 0.1483 | 0.9123 | 0.6707 | 0.8093 |
| (std) | 0.0003 | 0.0001 | 0.0005 | (0.0108) |
| ALMO | 0.1450 | 0.3043 | 0.4437 | 0.8716 |
| (std) | 0.0020 | 0.0000 | 0.0000 | (0.0021) |
| $\lambda$ | 0.0000 | 0.0000 | 0.0000 | - |