[Reviews · NeurIPS 2020]

Review 1

Summary and Contributions: This paper presents a new algorithm and theoretical analyses for learning problems that have multiple objectives or loss functions. For example, one might want to optimize both hinge loss and cross-entropy loss for robustness reasons, or optimize multiple population-based losses to ensure "AI fairness" across demographic groups. The contributions are (a) a formulation of the multi-objective problem in terms of worst-case linear combination of loss objectives, (b) Rademacher complexity analysis for the multi-objective case and a proof of Pareto optimality, and (c) an algorithm using stochastic mirror-prox gradient descent method and some promising results on MNIST, etc.

Strengths: Multi-objective learning problems are generally computationally expensive to solve; the paper's gradient descent style algorithm demonstrates a way to these problems in a more efficient and scalable fashion. The associated Rademacher complexity analysis and convergence guarantees further establishes the formulation on firm grounding. I am quite excited by the potential impact of this paper in making multi-objective learning more feasible.

Weaknesses: No major weakness. To add suggestions, I would be interested in seeing more experiments if the space allows (see below comments), but I think the amount of empirical work is comparable to other NeurIPS papers focusing on optimization algorithm/theory.

Correctness: Yes

Clarity: Yes, the paper is very clear.

Relation to Prior Work: Yes

Reproducibility: Yes

Additional Feedback: 1. The formulation of the (empirical) agnostic multi-objective loss objective (Eq 1 and 3) is based on a linear combination of objective. While you prove that the minimizer is Pareto optimal, it might be instructive to make explicit to the reader what kind of Pareto solution you achieve. For example, I am correct to believe that in Fig 1 (middle), the method achieves the Pareto solution indicated in the cross-hairs, but not any of the other Pareto solutions indicate in green? Also, it would be helpful to add discussion on what minimizers are achieved when you have convex vs nonconvex objective spaces. 2. Related to the above, can you clarify in Fig 2 whether you expect to achieve the similar lambda weights regardless of the initialization of parameter w_o and stochasticity in the method? 3. I didn't exactly follow how the AUC objective is computed in the experiments; I thought this is a ranking based loss, so do you assume a binary ranking since the experiments are classification problems? 4. Table 1: please be explicit in the caption whether these are train/dev/test loss. 5. The single-loss results in the Appendix are very interesting. If possible, it would be nice to squeeze a little more discussion about it into the main paper. == Response to author rebuttal == I appreciate the clear and concise response. It was informative and helpful. Again thanks for a nice paper! I learned something interesting and enjoyed reading it.


Review 2

Summary and Contributions: In this work the authors propose an agnostic learning framework for multi-objective problems. Most real world problems essentially require us to optimize multiple objectives, and there have been 2 primary directions of solution for the same, -- namely either weighted mixing of the objectives to achieve a scalar objective which is an extremely brittle approach in scenarios where it is difficult to determine the true mixing parameters or optimizing the whole pareto frontier which is not scalable. Thus the authors propose an agnostic learning strategy where it is not necessary to know or even estimate the mixing distribution. This is acheived by formulating a min-max optimization problem where they minimize the worst-case (max) expected parameterized loss which they solve using Stochastic Mirror Prox gradient ascent. Firstly they prove that the optimal solution generated by their procedure is actually a point on the pareto-optimal frontier, and secondly their evaluations on 3 well known domains prove their claim their framework learns the best using the worst case loss without ever actually selecting fixed point mixing weights.

Strengths: 1. This is a very important problem and its impact and and usefulness spans almost entire ML/AI community. Mixing distribution - agnostic learner is going to be extremely essential in formulating real-world problems which are implicitly multi-objective with unknown (or dynamic) mixing distributions. 2. The way the motivation section is written and each sub-problem highlighted and explained is very nice and is indeed a good example of how a paper should be written. For instance the paragraph Line 55-66 is very insightful about how mixing coefficients and their distribution is and one of the important factors in convergence to optimal solution. 3. The theoretical guarantee on how ALMO (proposed work) will actually find an optimal solution that is a point on the pareto-frontier is really a nice touch. This theoretically positions this work nicely . 4. The authors have kept the notations clear and concise and very legible. The trick of paramerized expression of the losses such that projections into the loss parameter space and the mixing parameter space can be shown jointly is a nice touch. 5. The broader impact section is well written and most relevant set of problems have been discussed.

Weaknesses: Just 3 concerns: 1. Equation 11 is expressed without the regularization terms ... and it is stated that extending it with the same is straightforward. Is that really so? It is not really that straightforward when projecting. SO authors must expand on this a bit. 2. Justification provided for not comparing against Pareto-frontier search methods is very loose and ambiguous. and not convincing in anyway. Authors should expand on this more. 3. This agnostic formulation in this case is related with solving a minmax problem ... in "robust MDPs" or even in adversarial games.... Authors should make an attempt to discuss the same. Finally, while the evaluations are acceptable they definitely have room for improvement, at least in the way the results are discussed. -- that is sub-par. Ideally the authors must define evaluation goals as in what is measured and for what reason.. what is it that they are trying to show. Then when discussed the tables/plots connect the discussion back to the goals.

Correctness: Most things appear correct.

Clarity: Yes the most of the paper has been written very well.. except some concerns in the experimental section ... which is mentioned in the weakness box.

Relation to Prior Work: Yes mostly good. One minor thing missing which has been highlighted in the weakness box.

Reproducibility: Yes

Additional Feedback: Authors may address the concerns mentioned in the weakness box.


Review 3

Summary and Contributions: The paper considers objective mixture problems and develops data-dependent generalization bounds for convex weightings of the loss. This bound is used to derive a training objective and optimization algorithm for the optimal loss weighting and hypothesis.

Strengths: The paper presents a principled approach to multi-objective loss optimization with the mixture setting nicely handled by considering an agnostic setting.

Weaknesses: The performance wrt/ mixture weights may be sensitive to the loss Lipschitz constants - it would have been nice to understand this sensitivity better. It isn't clear that re-normalizing losses so that they have the same scale is always appropriate. The experimental results in classification are lacking accuracy which is a standard metric. There are missing experimental details such as how long training was performed that do not provide confidence that the results could be reproduced. No real discussion of related work.

Correctness: The approach used in the learning guarantees appears reasonable, but I did not rigorously check.

Clarity: The paper is generally clear.

Relation to Prior Work: There is almost no discussion of related work.

Reproducibility: No

Additional Feedback: ### Regarding the rebuttal, I appreciate the extra information provided by the authors, and have upgraded my score. The relevant related work should be added to the final paper as well as a link to code. Also, my confusion about the accuracy metric was that it *was* included in the set of objective losses in the experiments - this is odd given that accuracy (0-1) is not Lipschitz. In other words, I was expecting to see it below the double horizontal lines of Tables 1 and 2, not above. The authors may want to address this subtlety. ###


Review 4

Summary and Contributions: The authors present a learning algorithm which automatically considers multiple loss functions using a minimax perspective. They provide theoretical analysis via Rademacher complexity bounds, motivating the final loss function's formulation. The authors demonstrate the utility of the method on 3 data sets.

Strengths: Weighting multiple learning objectives is a cumbersome challenge without additional machinery to help. The authors agnostic approach is well motivated with theoretical backing.

Weaknesses: The demonstration uses multiple classification losses. While each is different in their emphasis all of these losses are essentially aiming at the same high level goal. I should have liked to see at least one demonstration using fundamentally different objectives. For example, one might try to train an autoencoder for a multi-task problem, using the latent representation as the feature vector to solve the different tasks, which may be a mixture of types (e.g. classification and regression). In this set up, we have reconstruction loss for the autoencoder and a loss for each of the classification tasks. For something like this, it's not clear whether reconstruction is at all necessary or helpful, and one must contend with the different task-specific losses. Related work is not fully explored. In addition to the providing context for the presented work, there are several baseline methods that the authors ought to compare to. See my response in the prior work section below.

Correctness: The methods appear correct.

Clarity: The paper is very well written. The axis and title font sizes in Figure 1 are too small. Also, consider using both color and style to differentiate points for better representation in black/white printing.

Relation to Prior Work: Related work is not fully explored by the authors. For example, the authors cite Kendall et al. (2018), but do not discuss the method proposed in that work; learning loss weights with additional regularization. This is quite similar to the proposed work. I suggest the authors include the method of Kendal et al. as a baseline for comparison. Kendall, A., Gal, Y., & Cipolla, R. (2018). Multi-Task Learning Using Uncertainty to Weigh Losses for Scene Geometry and Semantics. Another potential baseline method is proposed in Chen, Z., Badrinarayanan, V., Lee, C.Y. and Rabinovich, A., (2018). Gradnorm: Gradient normalization for adaptive loss balancing in deep multitask networks. I'm sure there are others. Consider: Gong, T., Lee, T., Stephenson, C., Renduchintala, V., Padhy, S., Ndirango, A., Keskin, G. and Elibol, O.H., 2019. A comparison of loss weighting strategies for multi task learning in deep neural networks. Malkiel, I. and Wolf, L., 2020. MTAdam: Automatic Balancing of Multiple Training Loss Terms.

Reproducibility: Yes

Additional Feedback: 1. The authors did not seem to recognize the connection of their work to multi-task learning, and 2. The authors really need to include baselines from said field. In the rebuttal, the authors comment that “Agnostic Multi-Objective Learning” differs from “Agnostic Multi-Task Learning”. I don't see why this should be the case. I don't believe that multi-tasking is defined by the same loss function on different domains, even if this situation is common. I do acknowledge that some of the multi-task learning methods may not be directly comparable to the authors' methods as presented here. However, couldn't the authors' method be applied to multiple output spaces? If so, it seems to me that this is a missed opportunity as such an application would significantly increase the utility of the method.

[Author Response · NeurIPS 2020]

We thank all the reviewers for their feedback and pointers to relevant papers. We first address some general comments raised by multiple reviewers and next respond to individual questions.

**Extended discussion of related work:** We will add a detailed discussion of all the relevant papers mentioned by reviewers to the final version. This includes (Kendall et al., 2018), where they learn $1/\sigma$ of a Gaussian kernel as weights for base objectives, and (Chen et al., 2018), where they dynamically update gradient norms for each base loss. The number of empirical papers, such as those mentioned above, that explore training with multiple objectives/tasks is quite large. While we are happy to discuss as many of the relevant references as we can, the main focus of our paper, as concisely summarized by REV1, is to provide a theoretical analysis and motivation for the agnostic learning scenario.

**The goals and structure of our experiments:** As suggested by REV2, we will add a detailed formulation of the goals of the experiments and will connect them to the evaluation metrics we use. Experiments serve to 1) illustrate the application of our novel agnostic learning formulation; and 2) support a claim that since our algorithm optimizes for the worst case mixture, it provides more robust results than training with a fixed mixture. Thus, comparing ALMO with a uniform mixture baseline is the most direct and clean experiment to demonstrate 1) and 2). We use AUC, which is not included in the training loss mixture, as a robustness measure and show that ALMO has better AUC than the baselines.

**Connection to multi-task literature and baselines:** We recognize in the paper that multi-task learning is relevant, it is nevertheless distinct from our framework. That being said, the ALMO algorithm could indeed benefit from some of the multi-task literature; and as pointed out, in reverse, our algorithm could be relevant to that setting. The key distinction lies in the way we work with base loss functions and output spaces. While multi-tasking typically considers the same loss function on different domains where the distributions and output spaces may differ (and many methods derive from that assumption, e.g. Kendall et al. 2018), we consider different loss functions on the same output space. This difference is why we have limited the discussion of multi-task literature, although we did include some references. The topic of our paper is "Agnostic Multi-Objective Learning" which is a bit different from "Agnostic Multi-Task Learning".

There are specific reasons we did not use several multi-task learning algorithms mentioned by REV4 as baselines. First, Kendall et al. (2018) assumes that all base losses are applications of the same function (max likelihood in this case) on different output. This is where the scaling factor $1/\sigma$ comes from, which is treated as weights for base Gaussian likelihoods. We don't see how this method can be extended to our scenario where base losses do not necessarily represent Gaussian likelihoods and in fact, they come from different functional families, such as the hinge loss and cross-entropy loss. Moreover, our regularization admits a very different nature. Second, the GRADNORM of Chen et al. (2018) also applies the same loss to different outputs. Since losses have different magnitudes for different output domains, GRADNORM is a useful multi-task normalization method. It can be used as a helpful subroutine in ALMO to balance the gradients. However, directly normalizing the base losses was sufficient for our experiments. Finally, for several of the suggested multi-task methods, we are not sure they provide Pareto-optimal solutions.

REV1: The reviewer is correct about the type of the Pareto-optimal solution achieved by ALMO, we will include relevant details. We have verified that lambda weights (as in Figure 2) are not significantly different for different values of initial weights. As for the AUC objective, since we are working with multi-class classification (e.g., 10 classes for MNIST) the ranking is based on the relative probabilities of each class. As recommended by REV1, we will fix the caption in Table 1 (the figures are for the testing set), and will add a detailed discussion of single-loss results.

REV2: In the final version, we will add details of how to extend Equation 11 to the regularized problem: when a convex regularization term is added to the loss ($\mathcal{W}, \Lambda$ unchanged) the problem is still convex and the math is similar even for the projections. The minimax can indeed be viewed as an equilibrium in a two-player adversarial game, we will add more explanation for that. We provide more detail in lines 35-46 about why we don't apply Pareto-frontier search methods, but as suggested we will strengthen the argument and also illustrate with examples that in large-scale data settings, such as when the number of base losses is large, Pareto frontier estimation can often be computationally infeasible. On the other hand, our solution is theoretically proven to be Pareto-optimal, thus we don't need to verify that empirically.

REV3: The sensitivity to Lipschitz constants is directly captured by the $M_k$-weighted lambda norm and related terms in the generalization bound of Theorem 3. The zero-one loss that we report is 1–accuracy. As suggested, we will add more experimental details such as the training time (which took several hours at most). Please note that we have made sure all nontrivial and essential details for reproducibility are included in the NeurIPS submission.

REV4: Comments about related work and baselines are addressed above. Please note that Malkiel et al. was published only after the NeurIPS-2020 deadline. However, their optimization method can possibly help speed up the ALMO algorithm and we are interested in trying it. The auto-encoder experiment is definitely worth exploring in future work, though as we discussed, it is out of the scope of our learning scenario and the claims we put forward. Particularly, because there is no clear ground to assert that the goal of the learner is to optimize for the worst-case mixture of reconstruction, classification and regression losses. As hinted by REV4, the real goal of the learner might be to evaluate whether adding the reconstruction loss improves the generalization of the underlying model on multiple tasks.

[Meta-Review · NeurIPS 2020]

Four knowledgeable referees thoroughly evaluated the paper and author's response to their initial reviews. Three of them recommend acceptance, one reviewer leans towards marginal rejection. All things considered, I recommend accepting this paper as a poster. The reviewers insist that the more comprehensive comparisons relevant related work and a link to software code be included in the final revision. We strongly urge you to, and we trust that you will, make these changes.